# Repulsive Guidance Molecule-A as a Therapeutic Target Across Neurological Disorders: An Update

**DOI:** 10.3390/ijms26073221

**Published:** 2025-03-30

**Authors:** Vasilis-Spyridon Tseriotis, Andreas Liampas, Irene Zacharo Lazaridou, Sofia Karachrysafi, George D. Vavougios, Georgios M. Hadjigeorgiou, Theodora Papamitsou, Dimitrios Kouvelas, Marianthi Arnaoutoglou, Chryssa Pourzitaki, Theodoros Mavridis

**Affiliations:** 1Department of Neurology, Agios Pavlos General Hospital of Thessaloniki, 55134 Thessaloniki, Greece; 2Laboratory of Clinical Pharmacology, Aristotle University of Thessaloniki, 54124 Thessaloniki, Greece; dkouvelas@gmail.com (D.K.); chpour@gmail.com (C.P.); 3Department of Neurology, Nicosia General Hospital, Strovolos, 2031 Nicosia, Cyprus; liampasand@gmail.com (A.L.); vavougyios.georgios@ucy.ac.cy (G.D.V.); hadjigeorgiou.georgios@ucy.ac.cy (G.M.H.); 4Department of Neurology, Medical School, University of Cyprus, 1678 Nicosia, Cyprus; 5Department of Neurology, Democritus University of Thrace, 68100 Alexandroupolis, Greece; irenelazarid@gmail.com; 6Research Team “Histologistas”, Interinstitutional Postgraduate Program “Health and Environmental Factors”, Department of Medicine, Faculty of Health Sciences, Aristotle University of Thessaloniki, 54124 Thessaloniki, Greece; sofia_karachrysafi@outlook.com (S.K.); thpapami@auth.gr (T.P.); 7Laboratory of Histology-Embryology, Department of Medicine, Faculty of Health Sciences, Aristotle University of Thessaloniki, 54124 Thessaloniki, Greece; 8First Department of Neurology, AHEPA General Hospital of Thessaloniki, 54124 Thessaloniki, Greece; marnaoutoglou@yahoo.com; 9Department of Neurology, Tallaght University Hospital (TUH)/The Adelaide and Meath Hospital, Incorporating the National Children’s Hospital (AMNCH), D24 NR0A Dublin, Ireland; mavridismdr@gmail.com; 101st Neurology Department, Eginition Hospital, Medical School, National & Kapodistrian University of Athens, 11528 Athens, Greece

**Keywords:** repulsive guidance molecule-a, multiple sclerosis, neuromyelitis optica spectrum disorder, stroke, spinal cord injury, amyotrophic lateral sclerosis, epilepsy, auditory neuropathy

## Abstract

Repulsive guidance molecule-a (RGMa) has emerged as a significant therapeutic target in a variety of neurological disorders, including neurodegenerative diseases and acute conditions. This review comprehensively examines the multifaceted role of RGMa in central nervous system (CNS) pathologies such as Alzheimer’s disease, Parkinson’s disease, amyotrophic lateral sclerosis, multiple sclerosis, neuromyelitis optica spectrum disorder, spinal cord injury, stroke, vascular dementia, auditory neuropathy, and epilepsy. The mechanisms through which RGMa contributes to neuroinflammation, neuronal degeneration, and impaired axonal regeneration are herein discussed. Evidence from preclinical studies associate RGMa overexpression with negative outcomes, such as increased neuroinflammation and synaptic loss, while RGMa inhibition, particularly the use of agents like elezanumab, has shown promise in enhancing neuronal survival and functional recovery. RGMa’s responses concerning immunomodulation and neurogenesis highlight its potential as a therapeutic avenue. We emphasize RGMa’s critical role in CNS pathology and its potential to pave the way for innovative treatment strategies in neurological disorders. While preclinical findings are encouraging so far, further clinical trials are needed to validate the safety and efficacy of RGMa-targeted therapies.

## 1. Introduction

Repulsive guidance molecule-a (RGMa) is a glycosylphosphatidylinositol (GPI)-anchored membrane protein that belongs to the repulsive guidance molecule (RGM) family [1]. Originally identified for its role in axon guidance during embryonic development, RGMa is now recognized for its broader functions in the adult central nervous system (CNS), where it regulates neuroplasticity, axonal growth, myelination, and neuroinflammation [2]. RGMa operates through binding to its receptor, neogenin, which activates intracellular signaling pathways that inhibit neurite outgrowth, promote apoptosis, and disrupt the integrity of the blood–brain barrier [1,2].

Under normal physiological conditions, RGMa plays a role in maintaining neural network stability, but its expression is markedly upregulated in response to CNS injury or disease [3]. In neurological disorders, elevated RGMa levels contribute to the inhibition of axonal regeneration, demyelination, and neuronal damage [2]. For example, in multiple sclerosis (MS), RGMa exacerbates the neurodegenerative processes that lead to progressive disability [4]. In ischemic stroke, RGMa upregulation after injury contributes to neuronal death and limits recovery by preventing axonal growth [3]. Similarly, in spinal cord injury (SCI), RGMa impedes axon regeneration at the injury site, preventing functional recovery [5]. The growing body of evidence suggests that targeting RGMa with specific inhibitors, such as monoclonal antibodies [2,5], could offer therapeutic benefits across these and other neurological conditions, enhancing recovery and neuroprotection.

While previous literature reviews have mostly focused on extensively presenting the role of RGMa in CNS physiology and pathology [2], this comprehensive review aims to offer an update on the progress, novelties, and advancements regarding RGMa as a therapeutic target across neurological disorders. Through a narrative and critical approach, we herein gather and present recent updates about RGMa-targeted therapies for stroke, SCI, MS, NMOSD, neurodegenerative disorders, vascular dementia, auditory neuropathy, and epilepsy, aiming to inform and further guide future research, both basic and clinical. Evidence from included studies is summarized in Table 1 and extensively discussed below. Figure 1 depicts the main signaling pathways and mechanisms involved in RGMa pathological processes.

## 2. Stroke

In ischemic stroke, a pathophysiological cascade of complex inflammation processes takes place after the initial disruption of the blood flow. Soon after a stroke occurs, the blood–brain barrier (BBB) is impaired. Microglia swiftly relocate to the ischemic zones and make contact with the impaired endothelial cells of the BBB within the area of the occluded vessel [33,34,35]. Following a stroke, astrocytes transform into a reactive state, proliferate, and migrate toward the ischemic area where they form the glial scar, thus hindering axonal regeneration and neurogenesis [36].

Moreover, a change, partially induced by RGMa expression, in the inflammatory responses with a switch from microglia M2 (anti-inflammatory) to M1 (pro-inflammatory) polarization [37], along with the reactive astroglia formation [38] and BBB disruption [39], contributes to the secondary brain injury in the sequelae of ischemia/reperfusion after a stroke. In fact, prior studies have shown an increase in RGMa expression in various cell types within ischemic regions following cerebral ischemia/reperfusion, such as neurons, astrocytes, and endothelial cells [38,39].

The aforementioned render RGMa a compelling target for neuroprotection. Interestingly, studies have shown that the inhibition of RGMa reduces the ischemic stroke volume and improves functional outcomes [37]. In a more recent experimental stroke study, the researchers used a nanoplatform encapsulating an anti-RGMa monoclonal antibody, designed to target and neutralize RGMa [16]. This nanoplatform included microglia membrane coating for precise delivery to the ischemic regions. It was found that inhibition of RGMa was associated with significant reduction in infract volume and clinical improvement, offering a protective effect against ischemia/reperfusion injury. The anti-RGMa-loaded MiCM-NPs not only mitigated the harmful effects associated with its upregulation post-stroke but also enhanced the efficacy of stroke treatment by combining thrombus destruction with neuroprotective strategies [16]. This dual-function approach addresses both immediate and secondary damage mechanisms in ischemic stroke, potentially improving recovery outcomes.

The critical role of the neogenin and RGMa pathway in neuronal death following stroke has been proven with various experiments [15]. RGMa exerts its action by binding to the neogenin receptor, which in turn induces neuronal death when associated with lipid rafts. The pro-apoptotic function of neogenin depends mostly on its association with lipid rafts. In various in vitro and in vivo models (retinal ischemia and MCAO), inhibition of neogenin–lipid raft association significantly increased neuronal survival [15].

RGMa has also been investigated in neuronal metabolic reprogramming following ischemia-reperfusion (I/R) injury [17]. RGMa expression significantly increased after I/R, correlating with enhanced glycolytic metabolism in a middle cerebral artery occlusion/reperfusion mouse model. RGMa knockdown shifted neuronal energy metabolism toward oxidative phosphorylation and the pentose phosphate pathway, offering neuroprotection against I/R injury while downregulating phosphoglycerate kinase 1 expression, thereby reducing glycolytic flux. Additionally, PGK1 ubiquitination and degradation was promoted through disruption of USP10-PGK1 interaction, with findings suggesting the contribution of RGMa to neuronal damage through inhibition of USP10-mediated PGK1 degradation and glycolysis promotion [17].

Elezanumab, a neutralizing RGMa monoclonal antibody, has shown promising results by significantly improving neuromotor functions and reducing neuroinflammation in a middle cerebral artery occlusion (MCAO) experimental stroke model [15,18]. More specifically, when administered six hours post-MCAO, it promotes recovery of neuromotor function and reduces neuroinflammation and astrogliosis via influencing the activation and proliferation of microglia and astrocytes [29,38]. Similar outcomes were noted when the initial administration of elezanumab occurred 24 h post-MCAO, suggesting a potential neuroprotective effect, possibly allowing for a broader therapeutic time interval compared to existing treatment standards.

These findings underscore the potential of RGMa as a therapeutic target in stroke treatment, suggesting that RGMa inhibitors could play a crucial role in improving outcomes for stroke patients by mitigating neuronal damage and enhancing neuroplasticity and repair.

Currently, the EAISE study, a phase 2a, randomized, double-blind, placebo-controlled trial that aims to evaluate elezanumab in ischemic stroke, has been completed and is awaiting results [40]. In this study, patients with ischemic stroke were administered elezanumab or a placebo within 23 h of stroke onset and monitored for safety, efficacy, and outcomes related to neurological status and daily living activities in a 52-week follow-up period.

## 3. Spinal Cord Injury

RGMa significantly contributes to the inhibition of axonal growth and the preservation of an unfavorable environment following SCI. In animal experiments, RGMa was identified on the surface of activated microglia, which hinder recovery after acute axonal damage. Minocycline administration, a microglia activation inhibitor, was shown to reduce RGMa levels [41]. Similarly, inhibition of RGMa function with the use of monoclonal antibodies improved motor function, gait, and enhanced recovery on locomotor tests and behavioral assessments [19]. An increase in neuronal survival and plasticity of descending serotonergic pathways was noted, thus enabling corticospinal tract axonal regeneration. Even more interesting was that RGMa inhibition attenuated neuropathic pain responses, which correlated with decreased activation of microglia and a reduction in CGRP expression in the spinal cord [19]. This indicates the dual therapeutic potential of RGMa inhibition in both regenerative and pain management aspects of SCI treatment. Additionally, the delayed administration (24 h post-injury) of elezanumab following a thoracic SCI in rat models showed improvements in clinical/motor scores, underpinning the importance of expanding the therapeutic window for intervention post-SCI [20]. Anti-RGMa treatment demonstrated similar effect in cervical SCI models, which represent half of the SCIs in humans with detrimental neurological impairments and high mortality [21]. An interesting finding was also the faster recovery of spontaneous voiding ability and less post-trauma bladder wall hypertrophy, indicating functional improvements beyond motor recovery [20,21]. More evidence regarding elezanumab’s mechanism of action and potential originates from SCI studies of non-human primates [23]. Elezanumab mitigated the negative effect of RGMa following SCI by reducing its levels in the cerebrospinal fluid and decreasing its membrane-bound form around the lesion sites. Significant neuroplastic changes and sprouting of serotonergic fibers both rostral to the injury and within the ventral horn of lower thoracic regions were reported [23]. Moreover, as evidenced by anterograde tracing techniques, elezanumab appears to promote the density of corticospinal tract fibers, underpinning potential pathways for functional recovery. However, the therapeutic effectiveness of elezanumab appears highly dependent on the method of administration, as intrathecal infusion did not have the same effects as intravenous administration [23].

Similar to stroke, combination therapies have also been used in SCI [22,24]. The research conducted by Nakanishi et al. and Yamanaka et al. examined the synergic effect of anti-RGMa antibody treatment combined with repetitive transcranial magnetic stimulation (rTMS) for enhancing motor recovery after SCI in rats and monkeys, respectively [22,24]. Of importance is the timing of the combination therapy. Specifically, even though simultaneous application of rTMS and anti-RGMa antibody showed no significant benefit, a sequential treatment regimen where anti-RGMa antibody was followed by rTMS resulted in superior motor performance [22]. This sequential treatment enhanced the expression of Ca2+/calmodulin-dependent kinase II (CaMKII), which plays a role in neuroplasticity via improving neuronal connectivity [40]. Accordingly, the activation of the corticospinal tract with rTMS after its “rewiring” through anti-RGMa treatment created synergistic effects that expedited and enhanced motor recovery [24]. In another study of a monkey SCI model, anti-RGMa promoted the growth of corticospinal tract fibers into spinal cord regions, leading to improved manual dexterity. Additionally, after intracortical microstimulation, the contralesional motor cortex played a key role in recovery at later stages.

The ELASCIA study, a phase 2, double-blind, randomized clinical trial, is currently underway to investigate the therapeutic impact of elezanumab on patients with acute traumatic cervical spinal cord injury [42]. This study evaluates the efficacy of elezanumab versus a placebo administered within 24 h post-injury, focusing on the recovery of upper extremity motor functions over a period of 52 weeks [42].

## 4. Multiple Sclerosis

In MS, an autoimmune and autoinflammatory disorder of the CNS, demyelination leads to neurodegeneration and brain atrophy due to axonal damage. The neurodegenerative component of the disease has been linked with smoldering-associated worsening (SAW) [43] and progression independent of relapse activity (PIRA) [44] that are more prominent in the disease’s progressive forms, such as secondary progressive MS (SPMS). RGMa was found to be expressed by dendritic cells, and its receptor neogenin was found in T-cells [6,45]. Moreover, it was found to increase the adhesion of T-cells through activation of the GTPase Rap1 [6]. Dendritic cells transfected with RGMa siRNA had a lower capacity for EAE induction, while anti-RGMa neutralizing antibodies showed efficacy for the alleviation of clinical progression in experimental autoimmune encephalomyelitis (EAE) models of MS, reduction in infiltration of CNS by inflammatory cells, with attenuated demyelination and axonal loss, and reduced T-cell proliferation and cytokine release [6]. The role of RGMa in progressive MS and neurodegeneration was studied based on evidence from SCI studies, regarding the expression of RGMa on the surface of activated microglia [41]. Anti-RGMa antibodies mitigated the harmful effect of microglia on the regeneration of the axons [41], with reversal of inhibition of new neurite development in vitro, and in vivo findings such as promoted nerve fiber regeneration/remyelination, expedited functional recovery, and preserved retinal nerve fiber layer (RNFL), as examined with optical coherence tomography (OCT) [7]. After the identification of the RGMa gene as a possible modulator of the immune response in MS [46] and the hypothesis that it might contribute to cell death through interaction with neogenin, the presence of RGMa was reported in both active and chronic lesions in autopsy studies, as well as in normal-appearing white matter (NAWM) or gray matter (NAGM) [7]. Functional improvement observed in patients with progressive MS treated with intrathecal corticosteroid triamcinolone acetonide (TCA) [7] was correlated with decreased RGMa levels. Treatment with humanized anti-RGMa monoclonal antibody markedly inhibited secondary progression, inflammation, demyelination, and neurodegeneration in a mouse EAE model of SPMS while simultaneously enhancing motor functions in animals with spinal cord pathology [8]. Additionally, high-field magnetic resonance imaging (MRI) in anti-RGMa-treated mice with targeted EAE showed a repair-promoting effect on the blood–spinal cord barrier (BSCB), which was also supported by GeneChip and immunohistochemical analyses, with attenuation of the EAE-related vascular pathology [9]. In the same study, longitudinal MRI with concurrent dynamic contrast-enhanced (DCE) MRI and diffusion tensor imaging (DTI) analysis predicted later-phase demyelination, which was mitigated following anti-RGMa treatment, providing evidence for dynamic improvement of spinal cord pathology with the use of humanized anti-RMGa humanized antibody [9].

The anti-RGMa monoclonal antibody elezanumab (ABT-555) has been investigated in double-blind, placebo-controlled randomized studies. A phase 1 study evaluated its safety, tolerability, pharmacokinetics, and immunogenicity at a single ascending dose in healthy controls and at multiple ascending doses (150 mg, 600 mg, and 1800 mg) in MS patients, concluding in a prolonged half-life and a favorable pharmacokinetic profile of monthly or bimonthly doses of up to 1800 mg, as well as proportional increases in drug levels in the systemic circulation and cerebrospinal fluid (CSF) with escalating doses [10]. Importantly, no adverse events were identified. Reduced RGMa levels were mainly demonstrated in the patients’ CSF at the dose of 1800 mg intravenously, while an increase in IL-10, a cytokine with anti-inflammatory properties, was noted. However, no significant improvement was seen on brain MRI. In another report, headache was the most common adverse effect, and there were no significant differences in the patients’ clinical status [47]. Two phase 2 studies, RADIUS-R (208 participants) and RADIUS-P (123 participants), were completed [48], with elezanumab being well tolerated but not meeting efficacy endpoints, since no significant improvements in clinical or MRI outcomes were observed [11].

## 5. Neuromyelitis Optica Spectrum Disorder

Neuromyelitis optica spectrum disorder (NMOSD) is a rare immune-mediated disease of the CNS, primarily affecting the spinal cord and optic nerve, leading to severe attacks such as visual loss, paralysis, and neuropathic pain [49]. Without treatment, patients often develop significant long-term disability due to recurrent inflammatory episodes and incomplete recovery [50].

Several immunotherapies, including eculizumab, inebilizumab, satralizumab, and ravulizumab, have been approved for AQP4-IgG-positive NMOSD by regulatory agencies worldwide [51,52,53,54]. Despite these advances, alternative therapeutic targets remain of interest. RGMa has emerged as a promising candidate, with three animal studies demonstrating its potential benefits.

Harada et al. produced a rat model of NMO by injecting NMO-IgG into the spinal cord and then investigated if the blockage of RGMa could offer a beneficial effect in the treatment of the NMO-like disease [12]. It was observed that the administration of humanized anti-RGMa monoclonal antibodies (mAb) had a positive clinical and immunological impact in the NMO model rat. In particular, these rats had a delayed onset of symptoms and decreased severity of clinical signs 12 days following the administration of the anti-RGMa mAb. Furthermore, anti-RGMa mAb treatment led to reduced accumulation of activated microglia in NMO rats and decreased infiltration of IL-17A+ T-cells and axonal damage but preserved astrocytes, indicating an overall amelioration of immune response.

In an animal and post-mortem study, Iwamoto et al. found that RGMa-neogenin-induced neutrophil infiltration into the acute lesions drives the astrocytopathy by regulating CXCL2 expression in macrophages. Further, the authors showed that the RGMa-mAb had a beneficial effect on both AQP4-IgG-induced astrocytopathy and motor deficits, as well as neuropathic pain, in NMO rats [13]. In their study, Katsu et al. established a rat model of NMO by administering AQP4-Abs, and they investigated the possibility that the prophylactic administration of RGMa-Mab has a possible disease-modifying effect on blood–spinal cord barrier (BSCB) dysfunction [14]. Treatment with RGMa-Mab led to amelioration of both clinical symptoms and perivascular astrocytopathy of the spinal cord and immunosuppression by reducing the expression of proinflammatory cytokines/chemokines and the infiltration of inflammatory cells into the spinal cord. Interestingly, the analysis of CSF in rats treated with RGMa-mAb showed an improved CSF/serum albumin ratio and decreased AQP4-Abs influx.

## 6. Neurodegenerative Disorders

Synaptic loss seems to also contribute to Alzheimer’s disease pathology [55], and RGMa is present in amyloid plaques of Alzheimer’s disease patients [55]. A recent study investigated the role of total soluble RGMa as a potential neurodegenerative disease biomarker in human serum and CSF, using an ultra-performance liquid chromatography with tandem mass spectrometry method [56]. In the CSF of patients with mild cognitive impairment and Alzheimer’s disease, total soluble RGMa was twofold lower, and yet more data are required to establish a safe conclusion regarding the value of these results.

In Parkinson’s disease (PD), the gradual decline in motor function results from reduced dopamine production in the substantia nigra (SN) [27]. In this condition, RGMa expression is increased in the SN [57], and this overexpression causes loss of dopaminergic neurons and glial activation [58]. A recent study examined whether RGMa inhibition can alter the PD pathology in 1-methyl-4-phenyl-1,2,3,6-tetrahydropyridine (MPTP)-injected mice [27]. Firstly, MPTP treatment enhanced RGMa expression in the SN. Interestingly, both the intraventricular treatment with polyclonal anti-RGMa antibodies and the intravenous treatment with humanized monoclonal anti-RGMa antibodies ameliorated the decrease in dopaminergic neurons, as well as the microglia/macrophage activation induced by MPTP.

Furthermore, virus-mediated RGMa overexpression under the tyrosine hydroxylase promoter into the SN caused neuronal degeneration of the nigrostriatal pathway, accumulation of microglia/macrophages, and motor impairment. These results are consistent with a previous study that induced virus-mediated RGMa overexpression under the synapsin1 promoter [58].

RGMa overexpression also induced morphological changes in microglia. Minocycline treatment, which inhibits the activation of microglia [59], suppressed this morphological change and ameliorated RGMa-induced degeneration and motor deficits, suggesting that increased RGMa activates microglia with a possible involvement in its pathogenic effects.

In vitro experiments using cultured microglia were also performed, revealing that RGMa upregulated the genes related to pro-inflammatory pathways, such as tumor necrosis factor (Tnfα), suggesting that RGMa activates microglia directly.

Hence, targeting RGMa with antibodies could be a viable strategy to treat PD, along with other neurodegenerative diseases, where neurodegeneration and inflammation are key players.

Amyotrophic lateral sclerosis (ALS) is characterized by the progressive degeneration of motor neurons, and RGMa’s role in axonal growth inhibition and inflammation suggests it may contribute to disease progression [60].

Preclinical studies in ALS animal models have shown that elezanumab preserves motor neuron integrity and delays functional decline, promoting axonal repair and mitigating neuroinflammation. Specifically, anti-RGMa antibody treatment ameliorated the motor function and lifespan of mSOD1 mice [28]. In addition, RGMa concentration was elevated in the cerebrospinal fluid of 30 patients with ALS, as well as in transgenic mSOD1 mice [28].

Uncontrolled protein aggregation [61] and cytoskeletal dysregulation [62] are also involved in ALS pathology. Anti-RGMa antibody significantly decreased mutant SOD1 protein accumulation in motor neurons of mSOD1 mice via inhibition of actin depolymerization, suggesting a possible therapeutic mechanism [28]. While clinical trials are still forthcoming, the dual regenerative and immunomodulatory effects position elezanumab as a potential therapeutic option for ALS.

## 7. Vascular Dementia

A decline in hippocampal neurogenesis is linked to cognitive deterioration in conditions like Alzheimer’s disease [63,64], vascular dementia [65,66], and aging [67]. Modulating neurogenesis could offer therapeutic potential for neurodegenerative diseases [68].

Neogenin is expressed in newborn neurons of the dentate gyrus of the adult hippocampus [69], and its interaction with RGMa inhibits neurite outgrowth and induces cell death, hindering CNS regeneration [70]. A recent study explored RGMa’s role in vascular dementia using a bilateral common carotid artery stenosis (BCAS) mouse model [26]. Immunohistochemical analysis showed increased RGMa expression in the hippocampus, reduced neurogenesis, impaired cholinergic innervation, and cognitive decline. Treatment with an anti-RGMa antibody reversed these deficits, enhancing neurogenesis, promoting neuronal survival, and improving cholinergic innervation. Importantly, RGMa expression and antibody effects were observed only in the hippocampus, not in white matter lesions. These findings highlight RGMa’s role in vascular dementia and suggest that anti-RGMa antibodies could be a promising therapeutic strategy.

## 8. Auditory Neuropathy

Auditory neuropathy encompasses the loss of synapses between cochlear primary afferent neurons and sensory hair cells due to noise exposure or aging [71]. Both neogenin1 and RGMa are present at the synapse, and RGMa prevents regrowth and synaptogenesis of peripheral auditory nerve fibers with inner hair cells [25]. Nevoux et al. used both an in vitro model of kainic acid-induced excitotoxicity causing cochlear synaptopathy and an in vivo mouse model after noise-induced damage to the synapse [25]. Aiming to test whether RGMa inhibition would promote synapse regeneration, treatment with anti-RGMa antibody led to recovery from synaptic damage in the cochlea of newborn mice, while also leading to effective reversal of synaptopathy in adult mice. Administering anti-RGMa treatment one week after noise exposure restored cochlear afferent synapses and led to the recovery of auditory brainstem response wave-I amplitudes. Whether this reversal remains achievable at later time points requires further investigation. However, a potentially broad therapeutic window exists, as spiral ganglion neurons can survive long after the initial damage—months in animal models and even years in humans [25]. Thus, targeting RGMa with an antibody could represent a promising therapeutic approach for patients with auditory neuropathy.

## 9. Epilepsy

RGMa is implicated in epilepsy, with its downregulation linked to mossy fiber sprouting, neuroinflammation, and microglial activation [72,73]. Several studies have investigated RGMa as a therapeutic target. A study that included human and animal subjects demonstrated that overexpressing RGMa in the hippocampus via a lentiviral vector can suppress seizures by preventing mossy fiber sprouting [30]. Song et al. used pentylenetetrazol (PTZ) kindling to induce a temporal lobe epilepsy model, injecting recombinant RGMa protein and FAK inhibitor 14 intracerebroventricularly, reporting reduced FAK phosphorylation at Tyr397, seizure-suppressant effects, and RGMa-inhibited mossy fiber sprouting [31]. Finally, Feng et al. revealed that miR-20a-5p regulates RGMa and RhoA, influencing axonal growth and neuronal branching. Silencing miR-20a-5p in a pentylenetetrazol-induced epilepsy model prevents epileptogenesis by promoting RGMa-RhoA-mediated synaptic plasticity without affecting mossy fiber sprouting [32].

## 10. Discussion

RGMa has emerged as a promising therapeutic target across a wide range of neurological disorders, both neurodegenerative and acute conditions. This comprehensive review of RGMa’s role across various pathologies such as AD, PD, ALS, MS, NMOSD, SCI, stroke, vascular dementia, and auditory neuropathy highlights its diverse and critical impact on CNS pathology.

As an axon guidance molecule, RGMa plays a significant role in several biological processes, including cell proliferation, differentiation, adhesion, migration, neurogenesis, and synapse formation [6,38]. It also influences neuronal apoptosis [1,2], axon growth inhibition [41], and immune responses [12,46] through various signaling pathways, indicating its involvement in both normal CNS development and the pathological processes of numerous CNS diseases.

In neurodegenerative disorders like AD and PD, RGMa modulates neuroinflammation and neuronal survival, significantly impacting disease onset and progression [27,58]. Elevated RGMa expression has been associated with increased neuroinflammation, particularly concerning microglial activation [59]. Neutralizing RGMa, especially with agents such as elezanumab, has shown promise in alleviating neuroinflammation and enhancing functional outcomes in these models [27]. In ALS, RGMa neutralization provides hope by mitigating microglial and astrocytic activation, central components of the inflammatory cascade that accelerates neuronal loss [28]. RGMa’s role extends to epilepsy, where its downregulation has been linked to mossy fiber sprouting and neuroinflammation [30,31]. Targeting RGMa in this context may help suppress seizure activity and promote better outcomes for individuals with epilepsy. In MS and NMOSD, RGMa is implicated in neuroinflammation, demyelination, and axonal damage [6,45]. Studies exploring RGMa inhibition in MS and NMOSD animal models have revealed a reduction in inflammation severity, thus preventing further axonal loss and preserving myelination [6,7,8,9,45,46]. However, despite these promising findings, the efficacy of elezanumab in improving physical function in patients with MS remains inconclusive [32,47,48], necessitating further exploration in clinical trials involving NMOSD. The role of RGMa in SCI and stroke is particularly significant, as both conditions result in acute CNS injury marked by considerable neuroinflammation and tissue damage [11,12,19]. Elevated RGMa expression correlates with poor recovery and limited neuroplasticity following SCI [37], while RGMa neutralization has demonstrated improvements in motor function and neuroplasticity in animal models treated with elezanumab [22,23,24]. Similarly, early intervention with RGMa inhibitors post-stroke has been shown to enhance neuromotor function and minimize tissue damage, particularly when administered within a critical time window after injury [18]. In cases of vascular dementia, often stemming from chronic ischemic events, inhibiting RGMa signaling may enhance neurovascular repair mechanisms, potentially halting or slowing the progression of cognitive impairments associated with vascular disease [26]. Auditory neuropathy, while less studied, may benefit from targeting RGMa due to its involvement in neurodegeneration and neuronal survival within the cochlea. Given the molecular similarities between RGMa’s effects in the brain and other nervous system areas, its inhibition could preserve auditory function following nerve injury or degeneration [25].

In conclusion, RGMa represents a multifaceted therapeutic target across a broad spectrum of neurological diseases. Its involvement in neuroinflammation, neuronal survival, and axonal regeneration highlights its importance in both chronic neurodegenerative diseases and acute neurological injuries. The clinical potential of RGMa inhibitors such as elezanumab is promising, with preclinical studies demonstrating positive outcomes across various disorders. Nevertheless, the specific pathogenesis and signaling pathways of RGMa in CNS diseases remain incompletely understood, warranting further investigation through randomized controlled clinical trials. Such studies are essential for elucidating the mechanisms of RGMa and guiding its application in clinical treatment strategies for neurological disorders that currently lack effective therapies.

## Figures and Tables

**Figure 1 ijms-26-03221-f001:**
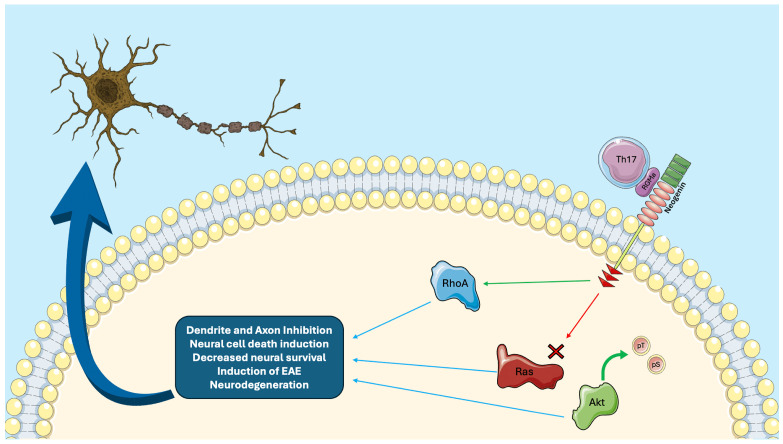
Overview of the mechanism of RGMa in the central nervous system. RGMa expressed in Th17 cells, via its interaction with neogenin, activates signaling pathways that inhibit axon growth and neuronal survival. Activation of RhoA, inactivation of Ras, and induction of Akt dephosphorylation contribute to the collapse of the growth cone and play an important role in axon guidance and regulation of neuronal death and survival. These signaling cascades play a crucial role in inflammation and angiogenesis, which are central to the pathophysiology of EAE. RGMa: repulsive guidance molecule-a, Th17: T-helper 17 cells, Akt: protein kinase B, RhoA: ras homolog gene family member A, EAE: experimental autoimmune encephalomyelitis, pT and pS: dephosporylation at sites Threonin-308 and Serine-473.

**Table 1 ijms-26-03221-t001:** Summary of studies investigating RGMa as a therapeutic target in neurological disorders.

Study	Intervention Studied	Neurological Disease	In Vitro Model	In Vivo Model	Important Findings
Muramatsu et al. 2011 [6]	RGMa-specific antibodies; RGMa siRNA	MS	BMDCs, CD4+ T-cells	MOG-induced EAE in naive C57BL/6 mice	RGMa activates Rap1, increasing CD4+ T-cell adhesion to ICAM-1. RGMa siRNA in dendritic cells reduces EAE induction. Anti-RGMa mAbs alleviate EAE symptoms, reducing CNS inflammation, demyelination, and axonal loss. Anti-RGMa mAbs suppress T-cell proliferation and pro-inflammatory cytokine release in mice and MS patient PBMCs.
Demincheva et al. 2015 [7]	5F9 rat RGMa mAbs; TCA intrathecally	MS	Human neuronal cell line NTera, human SH-SY5Y neuroblastoma cells	Autopsy tissues from progressive MS patients	RGMa is expressed in MS lesions, normal-appearing brain areas, and lymphocytes in perivenular infiltrates. Soluble RGMa decreased in CSF of MS patients, with improved function after TCA treatment. Neutralizing RGMa mAbs boosts regeneration, remyelination, and neuroprotection in animals.
Tanabe et al. 2018 [8]	Humanized anti-RGMa mAb	SPMS	NA	NOD-EAE mice	Anti-RGMa treatment significantly reduced disease progression, inflammation, demyelination, and axonal degeneration in SPMS model. Promoted growth of corticospinal tracts and motor recovery in targeted EAE mice with spinal cord lesions.
Hirata et al. 2022 [9]	Humanized anti-RGMa mAb	MS	NA	EAE	Anti-RGMa mAb improved BSCB repair and functional recovery in EAE mice. Treatment attenuated vascular pathology (endothelial thickening and collagen deposits). MRI predicted demyelination, with improved outcomes after treatment. GeneChip analysis confirmed the antibody’s effect on BSCB repair. Longitudinal MRI can serve as an imaging biomarker for evaluating drug effects in MS.
Kalluri et al. 2023 [10]	Elezanumab (ABT-555)—fully human mAb	MS	NA	MS patients, healthy participants	No significant drug-related adverse events. Tmax within 4 h of IV infusion, half-life ranging from 18.6 to 67.7 days. A percentage from 0.1% to 0.4% of elezanumab reached CSF, reducing free RGMa by >40%. Increased CSF levels of IL-10 (anti-inflammatory cytokine) with higher doses. Dosing of monthly or bimonthly administration, up to 1800 mg, with a potential loading dose of 3600 mg.
Cree et al. 2021(RADIUS-R and RADIUS-P) [11]	Elezanumab (ABT-555)—fully human mAb	MS	NA	RRMS, SPMS patients (phase 2 clinical trials)	Primary endpoint (Overall Response Score) was not met in either study. No clinically significant changes in secondary measures (SDMT, LCVA, MFIS-5). No positive effect on MRI brain and spinal cord assessments. Elezanumab was well tolerated with no safety risks. Severe adverse events were similar across groups.
Harada et al. 2018 [12]	Humanized anti-RGMa mAb	NMOSD AQP4-IgG	NA	Murine NMOSD model induced by AQP4-IgG infusion (Wistar rats)	Treatment with anti-RGMa mAb delays the onset and ameliorates the severity of clinical symptoms of NMOSD in model rats. Anti-RGMa mAb partially restores AQP4 and GFAP expression in NMOSD model rats. Anti-RGMa mAb reduces immune responses in NMO rats. RGMa inhibition attenuates the infiltration of IL-17A+ T-cells. RGMa inhibition attenuates neuronal damage in NMOSD model rats.
Iwamoto et al. 2022 [13]	Humanized anti-RGMa mAb	NMOSD AQP4-IgG	Cultured macrophages stained with anti-Iba1 or anti-neogenin to confirm Iba1 and neogenin expression in cultured peritoneal macrophages	Murine NMOSD model induced by MBP immunization and a single intraperitoneal injection of AQP4-IgG (Female Lewis rats, 8–12 weeks old)	The expression of RGMa is demonstrated in spared neurons and astrocytes, while its receptor neogenin is expressed by infiltrating macrophages. AQP4-IgG-induced astrocytopathy and clinical exacerbation in NMOSD rats is reduced by anti-RGMa mAb treatment. The administration of RGMa-mAb leads to a significant suppression of neutrophil infiltration and a reduced expression of neutrophil chemoattractants. RGMa directly mediates the expression of CXCL2 in macrophages. Treatment with RGMa-mAb can effectively improve the NMOSD-associated neuropathic pain.
Katsu et al. 2024 [14]	Humanized anti-RGMa mAb	NMOSD AQP4-IgG	NA	Murine NMOSD model induced by MBP immunization and AQP4-IgG infusion (female Lewis rats (LEW/CrlCrlj), 7 weeks old)	Anti-RGMa mAb reduces both the severity of perivascular astrocytopathy in the spinal cord and the clinical symptoms. Anti-RGMa mAb suppresses the expression of proinflammatory cytokines/chemokines and the infiltration of inflammatory cells into the spinal cord. Anti-RGMa mAb treatment improves the CSF/serum albumin ratio and reduces AQP4-Abs influx. Anti-RGMa mAb is a potential therapeutic option for BSCB dysfunction associated with NMOSD.
Shabanzadeh et al. 2015 [15]	AE12-1Y—fully human mAb	Stroke	Sprague–Dawley rat retinal whole-mount cultures; mouse E16 cortical neurons	eMCAO in Sprague–Dawley rats	Anti-RGMa mAb prevents neogenin’s association with lipid rafts, thus protecting from CNS ischemic tissue damage. Anti-RGMa mAb significantly improves functional recovery when administered up to 6 h after artery occlusion. Anti-RGMa mAb enhances the complexity of the neuronal network following ischemia.
Cheng et al. 2024 [16]	MiCM-coated and LIFU/magnetic responsive nanoparticle loaded with anti-RGMa mAb	Stroke	Mouse brain microvascular endothelial cells (bEnd.3 cell), isolated from brain tissue deriving from a mouse with endothelioma	LCCA embolization model in Sprague–Dawley rats; eMCAO model in C57 mice	MiCM nanoparticles facilitated targeted delivery to ischemia-damaged endothelial cells, enhancing the recanalization of occluded blood vessels. Anti-RGMa mAb provided neuroprotection by mitigating ischemia/reperfusion injury, significantly improving the functional recovery and reducing infarct volume in vivo. The nanoparticle design allowed for the visualization of thrombi using ultrasound/photoacoustic imaging techniques, improving the diagnosis and treatment monitoring of ischemic strokes.
Wang et al. 2024 [17]	RGMa knockdown (AAV-shRGMA); PKG1 overexpression (AAV-PGK1); in vitro manipulations (siRGMA, siUSP10, PGK1, USP10 overexpression, LV-RGMA)	Stroke	OGD/R model constructed using primary cortical neurons isolated from fetal rats	MCAO/R mouse model (C57BL/6J male mice, 8–10 weeks old)	RGMa expression increases after I/R in vivo and in vitro. MCAO/R mice show elevated glycolytic metabolic products and increased expression of glycolytic pathway proteins compared to controls. RGMa is linked to neuronal energy metabolism, influencing the balance between glycolysis and oxidative phosphorylation. RGMa knockdown shifts metabolism towards oxidative phosphorylation, reducing ischemic reperfusion injury. RGMa knockdown downregulates PGK1 expression, with a decrease in glycolytic flux after I/R. RGMa regulates PGK1 degradation through USP10, reduces USP10-PGK1 interaction, and enhances PGK1 ubiquitination. Mechanistically, RGMa may contribute to neuronal damage by inhibiting USP10-mediated PGK1 degradation, promoting glycolysis under ischemic conditions. RGMa is a potential therapeutic target for mitigating neuronal injury after I/R through modulation of energy metabolism.
Jacobson et al. 2024 [18]	Elezanumab (ABT-555)—fully human mAb	Stroke	NA	pMCAO model in male New Zealand White rabbits	Elezanumab treatment significantly improved neuromotor function, demonstrating effectiveness even when the first dose was administered 6 h after the stroke. Decreased microglial and astrocyte activation suggested that neutralizing RGMa can effectively reduce neuroinflammation associated with ischemic stroke.
Mothe et al. 2017 [19]	AE12-1 and AE12-1Y human anti-RGMa mAb	SCI	E18 mouse cortical neurons	Clip impact-compression T8 thoracic injury in Wistar rats	Treatment with human anti-RGMa mAbs resulted in significant improvement in motor function and promoted axonal regeneration. Inhibiting RGMa promoted preservation of neurons around the injury site, reducing neuronal death. Anti-RGMa mAbs increased plasticity of serotonergic pathways, enhancing the body’s innate mechanisms for nerve recovery. Anti-RGMa mAbs decreased neuropathic pain via reducing activation of pain-associated microglia in the spinal cord.
Mothe et al. 2020 [20]	Elezanumab (ABT-555)—fully human mAb	SCI	NA	Clip impact-compression T8 thoracic injury in Wistar rats	Delayed elezanumab administration (up to 24 h post-injury) showed significant improvement in motor function, gait, and spontaneous voiding ability. Increased neuronal survival and axonal plasticity around the lesion site. Reduction in lesion volumes and increase in the number of surviving neurons and serotonergic fibers around the injured area. Earlier recovery of spontaneous voiding and less bladder wall hypertrophy compared to untreated controls.
Mothe et al. 2022 [21]	Elezanumab (ABT-555)—fully human mAb	SCI	NA	Clip impact-compression C6/7 cervical injury in Wistar rats	Delayed (up to 24 h) elezanumab administration resulted in increased neuronal survival and axonal sprouting near the injury site. Improvements in motor function tests and earlier recovery of spontaneous voiding ability.
Nakanishi et al. 2019 [22]	Anti-RGMa antibody (rat IgG; IBL) + rTMS	SCI	NA	Dorsal hemisection of spinal cord at T8 in C57BL/6 J rats	Sequential treatment with anti-RGMa antibody followed by rTMS significantly enhanced motor recovery in mice with spinal cord injuries. Sequential treatment notably increased the expression of CaMKII in the motor cortex, which is associated with neural plasticity and long-term potentiation.
Nakagawa et al. 2019 [5]	Anti-RGMa antibody + ICMS	SCI	NA	Hemisection of spinal cord at C6/7 cervical level in rhesus monkeys (Macaca mulatta, 3–5 years old, 3.8–5.4 kg)	RGMa inhibition promoted CST fiber growth into key spinal cord regions (laminae VII and IX). Manual dexterity was restored following anti-RGMa antibody. Contralesional motor cortex was critical for recovery at later stages.
Jacobson et al. 2021 [23]	Elezanumab (ABT-555)—fully human mAb	SCI	NA	T9/10 hemicompression in African green monkeys (Chlorocebus sabaeus)	Elezanumab treatment led to notable enhancements in locomotor function only after systemic intravenous administration. MRI analysis showed improvement of the microstructural integrity of extralesional tissue. Increase in the density of corticospinal tract fibers and significant sprouting of serotonergic fibers, particularly around the lesion site and in the ventral horn of lower thoracic regions.
Yamanaka et al. 2021 [24]	Anti-RGMa antibody + rTMS	SCI	NA	Hemisection of spinal cord at C6/7 cervical level in Japanese monkeys (adult male Macaca fuscata monkeys, 5.4–7.1 kg)	The combination of anti-RGMa antibody treatment followed rTMS significantly enhanced the recovery of dexterous hand movements in primates with spinal cord injuries. Combination therapy resulted in the rewiring of corticospinal tract fibers.
Nevoux J et al. 2021 [25]	Anti-human RGMa antibody	Auditory neuropathy	Kainate excitotoxicity mouse model with cochlear culture from postnatal day 4 to 6 (adapted model of excitatory cochlear synaptopathy)	CBA/CaJ male mouse model of noise-induced synapse damage	Both neogenin1 and RGMa are present at the cochlear synapse. Treatment with anti-RGMa antibody led to regeneration of synapses both in vitro and in vivo.
Yamamoto M et al. 2024 [26]	Anti-RGMa neutralizing antibody	Vascular dementia	NA	Bilateral common carotid artery stenosis mouse model of vascular dementia	Immunohistochemical analysis revealed that BCAS mice exhibited increased RGMa expression in the hippocampus, which coincided with reduced neurogenesis and impaired cholinergic innervation. Anti-RGMa antibody treatment reversed these pathological changes and cognitive deficits. Anti-RGMa antibody treatment promoted the survival and maturation of newborn neurons. Anti-RGMa antibody treatment significantly ameliorated the impaired cholinergic innervations. RGMa expression and the anti-RGMa antibody effects were observed only in the hippocampus, not in white matter lesions.
Oda W et al. 2021 [27]	Humanized anti-RGMa antibody, polyclonal anti-RGMa antibodies	PD	Coculture of microglia from embryonic mice cortex and RGM-expressing CHO cells	MPTP mouse model of PD (adult male C57BL/6J, 8 weeks old); mouse model with AAV-mediated RGMa overexpression in SN	RGMa expression is increased in the SN of MPTP-treated mice. Anti-RGMa antibodies reduced the loss of dopaminergic neurons and the microglia/macrophage activation in the SN of MPTP-treated mice. Selective expression of RGMa in dopaminergic neurons in the SN induced neuronal loss/degeneration and inflammation, resulting in a progressive movement disorder. The pathogenic effects of RGMa overexpression were attenuated by treatment with minocycline, which inhibits microglia and macrophage activation. Increased RGMa expression upregulated pro-inflammatory cytokine expression in microglia.
Shimizu M et al. 2023 [28]	Humanized anti-RGMa mAb	ALS	Evaluation of the uptake of recombinant soluble SOD1 mutations on rats’ primary cortical neurons	Transgenic mice overexpressing the mutant human superoxide dismutase1 (mSOD1 mice); ALS patients (Awaji criteria)	RGMa concentration was elevated in the CSF of ALS patients and transgenic mSOD1 mice. Anti-RGMa antibody treatment ameliorated motor function and lifespan in mSOD1 mice. RGMa/Neogenin1 signaling promoted the cellular uptake of the SOD1 protein in vitro. Anti-RGMa antibody significantly decreased mutant SOD1 protein accumulation in motor neurons of mSOD1 mice via inhibition of actin depolymerization. Anti-RGMa antibody inhibited cellular uptake of mutant SOD1 protein, possibly by reinforcing the neuronal actin barrier.
Huang et al. 2021 [29]	Elezanumab (ABT-555)—fully human mAb	Optic nerve injury; optic neuritis; MS; demyelination	BMP competition; ELISA; BMP reporter gene assay; RGMa repulsive activity assay (human neuroblastoma SH-SY5Y cells)	Optic nerve crush model in rats; optic neuritis model in rats; spinal-targeted EAE rat model; mouse cuprizone model; iron metabolism study on female SD rats; pharmacokinetics and bioanalysis in SD rats and cynomolgus monkeys	Elezanumab competes with ABT-207 in binding to N-RGMa, but lacks RGMc cross-reactivity, having no adverse effect on iron metabolism. Elezanumab neutralizes repulsive activity of soluble RGMa in vitro and inhibits membrane RGMa mediated BMP signaling. In optic nerve crush and optic neuritis models, elezanumab stimulated axonal regeneration and prevented retinal nerve fiber layer degeneration. In a spinal-targeted experimental autoimmune encephalomyelitis (EAE) model, elezanumab stimulated axonal regeneration and remyelination, reduced inflammatory lesion area, and ameliorated functional recovery. In a mouse cuprizone model, elezanumab reduced demyelination.
Chen et al. 2017 [30]	Lentivirus-mediated RGMa overexpression in the hippocampus of epileptic animal models	Epilepsy	Patch–clamp recordings from CA1 pyramidal neurons (acute slice model of epileptiform activity)	Pilocarpine-induced rat model (male Sprague–Dawley rats, weight: 200–230 g); PTZ kindling model (male Sprague–Dawley rats, weight: 200–230 g); TLE patients	RGMa expression was significantly reduced in TLE patients and epileptic rat models. Lentivirus-mediated overexpression of RGMa in the hippocampus reduced spontaneous seizures. Seizure suppression was linked to reduced mossy fiber sprouting in the hippocampus. RGMa overexpression inhibited neuronal hyperexcitability by suppressing NMDAR-mediated currents.
Song et al. 2019 [31]	Intracerebroventricular injection of recombinant RGMa protein and FAK inhibitor 14	Epilepsy	NA	PTZ kindling model (male Sprague–Dawley rats, weight: 120–180 g, age: 40–45 days old)	RGMa protein and FAK inhibitor 14 reduced seizures in the PTZ model. RGMa protein decreased FAK (Tyr397) phosphorylation, while FAK inhibitor 14 reduced FAK-p120GAP interaction and Ras expression. RGMa suppressed mossy fiber sprouting, but FAK inhibitor 14 did not. Findings suggest RGMa as a potential epilepsy treatment, acting through the FAK-p120GAP-Ras signaling pathway.
Feng et al. 2020 [32]	miR-20a-5p silencing	Epilepsy	293T cells, PC12 cells and primary hippocampal neurons	PTZ kindling model (Sprague–Dawley rats, weight: 200–220 g, age: 7–9 weeks old and post-natal day 1)	miR-20a-5p regulates RGMa, influencing RhoA in the context of epileptogenesis. Silencing miR-20a-5p prevents epileptogenesis through RGMa-RhoA-mediated synaptic plasticity. miR-20a-5p-RGMa-RhoA pathway affects axonal growth and neuronal branching in vitro. Silencing miR-20a-5p in the in PTZ epilepsy model did not change mossy fiber sprouting but inhibited epileptogenesis.

RGMa: Repulsive guidance molecule-a; siRNA: small interfering RNA; MS: multiple sclerosis; SPMS: secondary progressive multiple sclerosis; RRMS: relapsing-remitting multiple sclerosis; NMOSD: neuromyelitis optica spectrum disorder; BMDCs: bone marrow-derived dendritic cells; MOG: myelin oligodendrocyte glycoprotein; EAE: experimental autoimmune encephalomyelitis; TCA: triamcinolone acetonide; NOD mice: nonobese diabetic mice; NA: not applicable; BSCB: blood–spinal cord barrier; CSF: cerebrospinal fluid; MiCM: microglia membrane; LIFU: low-intensity focused ultrasound; MRI: magnetic resonance imaging; LCCA: left common carotid artery; eMCAO: embolic middle cerebral artery occlusion; pMCAO: permanent middle cerebral artery occlusion; rTMS: repetitive transcranial magnetic stimulation; SCI: spinal cord injury; PD: Parkinson’s disease; ALS: amyotrophic lateral sclerosis; mab/mabs: monoclonal antibody/antibodies; AAV: adeno-associated virus; SN: substantia nigra; BCAS: bilateral common carotid artery stenosis; mSOD1: mutant human superoxide dismutase1; SDMT: symbol digit modalities test; LCVA: low-contrast visual acuity; MFIS-5: Modified Fatigue Impact Scale 5; CaMKII: Ca2+/calmodulin-dependent kinase II; MPTP: 1-methyl-4-phenyl-1,2,3,6-tetrahydropyridine; SOD1: superoxide dismutase 1; AQP4-IgG: aquaporin-4 immunoglobulin G; SD: Sprague–Dawley; OGD/R: oxygen–glucose deprivation/reoxygenation; MCAO/R: middle cerebral artery occlusion/reperfusion; I/R: ischemia-reperfusion; PGK1: phosphoglycerate kinase 1; ICMS: intracortical microstimulation; CST: corticospinal tract; PTZ: pentylenetetrazol; TLE: temporal lobe epilepsy.

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
