# Peer review of "Repulsive Guidance Molecule-A as a Therapeutic Target Across Neurological Disorders: An Update"

_ijms, 2025, doi:10.3390/ijms26073221_

Round 1
Reviewer 1 Report
Comments and Suggestions for Authors
This review comprehensively examines the multifaceted role of RGMa in central nervous system (CNS) pathologies such as Alzheimer’s disease, Parkinson’s disease, amyotrophic lateral sclerosis, multiple sclerosis, neuromyelitis optica spectrum disorder, spinal cord injury, stroke, vascular dementia, auditory neuropathy, and epilepsy. The mechanisms through which RGMa contributes to neuroinflammation, neuronal degeneration, and impaired axonal regeneration are herein discussed. The authors emphasize RGMa’s critical role in CNS pathology and its potential to pave the way for innovative treatment strategies in neurological disorders. Before this article can be officially published, the following issues need to be addressed:
- I note that a previously published review article on a related topic is already available in the literature (DOI: 10.1155/2021/5532116). While the authors have cited this work in the main text, it would be imperative for them to clearly delineate the distinctions between their current research contributions and those documented in the aforementioned review. Specifically, the manuscript should explicitly address the novel aspects, methodological advancements, or substantive extensions that differentiate this work from the previously reported comprehensive analysis.
- For a review article, the inclusion of appropriate figures and schematics is essential to enhance readability and conceptual clarity. It is recommended that the authors incorporate illustrative diagrams, such as a schematic summarizing the mechanisms underlying RGMa-induced neuroinflammation, neurodegeneration, and impaired axonal regeneration. These visual aids would strengthen the synthesis of complex pathways and provide readers with a more intuitive understanding of the discussed pathological processes.
Author Response
We would like to thank Reviewer 1 for their accurate comments.
Comment 1:
"I note that a previously published review article on a related topic is already available in the literature (DOI: 10.1155/2021/5532116). While the authors have cited this work in the main text, it would be imperative for them to clearly delineate the distinctions between their current research contributions and those documented in the aforementioned review. Specifically, the manuscript should explicitly address the novel aspects, methodological advancements, or substantive extensions that differentiate this work from the previously reported comprehensive analysis."
Response 1:
We absolutely agree that differentiation from previous similar reviews is needed and we thank the reviewer for this comment. We have revised the manuscript's introduction accordingly:
"While previous literature reviews have mostly focused in extensively presenting the role of RGMa in CNS physiology and pathology (2), this comprehensive review aims to offer an update on the progress, novelties and advancements regarding RGMa as a therapeutic target across neurological disorders. Through a narrative and critical approach, we herein gather and present recent updates about RGMa-targeted therapies for stroke, SCI, MS, NMOSD, neurodegenerative disorders, vascular dementia, auditory neuropathy, and epilepsy, aiming to inform and further guide future research, both basic and clinical. Evidence from included studies is summarized in Table 1 and extensively discussed below."
[page. 4-5, last paragraph of "Introduction"]
Comment 2:
For a review article, the inclusion of appropriate figures and schematics is essential to enhance readability and conceptual clarity. It is recommended that the authors incorporate illustrative diagrams, such as a schematic summarizing the mechanisms underlying RGMa-induced neuroinflammation, neurodegeneration, and impaired axonal regeneration. These visual aids would strengthen the synthesis of complex pathways and provide readers with a more intuitive understanding of the discussed pathological processes.
Response 2:
We appreciate and thank the reviewer for the comment. Despite mechanisms of action not consisting the primary aim of this study, inclusion of a schematic would indeed increase readability. In our revised manuscript Figure 1 concisely depicts the main signalling cascades discussed in the paper in terms of neurodegeneration, impaired axonal regeneration and neuroinflammation.
Reviewer 2 Report
Comments and Suggestions for Authors
The author's provide a comprehensive review of repulsive guidance molecule-A in the context of numerous diseases, including Alzheimer's, Parkinson's, ALS, MS, spinal cord injuries, epilepsy and others. The review is organized into two sections, one is a table listing references and their important findings while the second summarizes what studies involving RGMa in the context of diseases. This format, particularly the table, is helpful.
Minor concerns:
line 122-133 RGMA is used whereas in other sections RGMa is used as the abbreviation.
Author Response
We would like to thank the reviewer for their time and comments.
Comment 1: line 122-133 RGMA is used whereas in other sections RGMa is used as the abbreviation
Response 1: We have fixed all "RGMa" abbreviations throughout the manuscript